# Resveratrol and Quercetin Administration Improves Antioxidant DEFENSES and reduces Fatty Liver in Metabolic Syndrome Rats

**DOI:** 10.3390/molecules24071297

**Published:** 2019-04-03

**Authors:** Maria Esther Rubio-Ruiz, Verónica Guarner-Lans, Agustina Cano-Martínez, Eulises Díaz-Díaz, Linaloe Manzano-Pech, Anel Gamas-Magaña, Vicente Castrejón-Tellez, Concepción Tapia-Cortina, Israel Pérez-Torres

**Affiliations:** 1Department of Physiology, Instituto Nacional de Cardiología “Ignacio Chávez”, Juan Badiano 1, Sección XVI, Tlalpan, Mexico City 14080, Mexico; esther_rubio_ruiz@yahoo.com (M.E.R.-R.); gualanv@yahoo.com (V.G.-L.); cmamx2002@yahoo.com.mx (A.C.-M.); anel.gama@hotmail.com (A.G.-M.); vcastrejn@yahoo.com.mx (V.C.-T.); 2Department of Reproductive Biology, Instituto Nacional de Ciencias Médicas y Nutrición “Salvador Zubirán”, Vasco de Quiroga 15, Sección XVI, Tlalpan, Mexico City 14000, Mexico; eulisesd@yahoo.com; 3Department of Pathology, Instituto Nacional de Cardiología “Ignacio Chávez”, Juan Badiano 1, Sección XVI, Tlalpan, Mexico City 14080, Mexico; pertorisr@yahoo.com.mx; 4Colegio de Ciencias y Humanidades. Licenciatura en Promoción de la Salud. Academia de salud comunitaria. Universidad Autónoma de la Ciudad de México; Plantel San Lorenzo Tezonco, Mexico City 06720, Mexico; tapiacc@hotmail.com

**Keywords:** resveratrol, quercetin, metabolic syndrome, fatty liver, oxidative stress

## Abstract

Mixtures of resveratrol (RSV) + quercetin (QRC) have antioxidant properties that probably impact on fatty liver in metabolic syndrome (MS) individuals. Here, we study the effects of a mixture of RSV + QRC on oxidative stress (OS) and fatty liver in a rat model of MS. Weanling male Wistar rats were separated into four groups (*n* = 8): MS rats with 30% sucrose in drinking water plus RSV + QRC (50 and 0.95 mg/kg/day, respectively), MS rats without treatment, control rats (C), and C rats plus RSV + QRC. MS rats had increased systolic blood pressure, triglycerides, insulin levels, insulin resistance index homeostasis model (HOMA), adiponectin, and leptin. The RSV + QRC mixture compensated these variables to C values (*p* < 0.01) in MS rats. Lipid peroxidation and carbonylation were increased in MS. Total antioxidant capacity and glutathione (GSH) were decreased in MS and compensated in MS plus RVS + QRC rats. Catalase, superoxide dismutase isoforms, peroxidases, glutathione-S-transferase, glutathione reductase, and the expression of Nrf2 were decreased in MS and reversed in MS plus RVS + QRC rats (*p* < 0.01). In conclusion, the mixture of RSV + QRC has benefic effects on OS in fatty liver in the MS rats through the improvement of the antioxidant capacity and by the over-expression of the master factor Nrf2, which increases the antioxidant enzymes and GSH recycling.

## 1. Introduction

Oxidative stress (OS) is an imbalance between pro-oxidant agents that are elevated, such as super oxide (O_2_^−^) and hydrogen peroxide (H2O2), and the antioxidant systems that are depleted, including the enzymatic and non-enzymatic defenses [1]. Antioxidant substances delay the oxidation process, inhibiting the damaging chain that is initiated by free radicals and the subsequent oxidizing reactions [2]. Diets that are rich in carbohydrates and/or saturated fatty acids increase oxidative damage in the body and some studies have shown the importance of reactive oxygen species (ROS) in the development and progression of cardiovascular diseases, such as obesity and diabetes; hence, antioxidants are necessary to counteract the chronic oxidative effects [3,4].

Traditional medicine and a growing number of reports in the literature suggest the importance of natural antioxidants, such as polyphenols and flavonoids extracted from vegetables and fruits to treat many diseases. Polyphenols and flavonoids exhibit a wide range of biological and pharmacological activities both in vivo and in vitro; they reduce cellular oxidative damage and possess anti-inflammatory and antiatherogenic properties. Their effects include liver protection [2,5].

Resveratrol (3,4′,5-trans-trihydroxystilbene, RSV) is a natural polyphenol with antioxidant, anti-inflammatory, and anti-cancer properties, which is found in red wine, grapes, berries, and peanuts. It stimulates sirtuin (SIRT) expression. SIRT1 and 3 promote the synthesis of antioxidant enzymes in the liver. Some studies have also demonstrated that RSV modulates lipid and lipoprotein metabolisms [6]. Additionally, supplemental doses of RSV prevent nearly all of the detrimental changes due to high-calorie intake. Furthermore, polyphenolic substances increase the metabolic rate and fat oxidation, implying their potential use as anti-obesity treatments [6,7]. Quercetin (QRC) is a flavonoid that is present in onions, apples, and other fruits, and it can favor protection from OS and chronic diseases, including neurodegenerative and cardiovascular disorders, inflammation, and cancer [8].

The signs that are involved in the metabolic syndrome (MS) are among the diseases that might benefit from the antioxidant therapy with RSV and QRC [6,9]. The association of at least three of the following signs may define the MS: hypertension, dyslipidemia, insulin resistance (IR), and obesity. In our laboratory, we have developed a MS rat model that is induced by the chronic administration of 30% sucrose in the drinking water for 20–24 weeks since weaning. The physiological abnormalities that are usually present in this model are high systolic blood pressure (SBP), hypertriglyceridemia, obesity, hyperinsulinemia, and IR. There is also an increase in the serum concentration of free fatty acids that may accumulate as triglycerides (TG) in hepatocytes generating nonalcoholic fatty liver (NAFL). It has been reported that 70% of the adult patients and 25.5% of the pediatric patients with MS have NAFL [10]. Liver fatty acid accumulation could produce an excess of reactive oxygen species (ROS) and the loss of the antioxidant hepatic reserves. A decrease in glutathione (GSH) concentration, low activities of superoxide dismutase (SOD) and isoforms and catalase (CAT), and a decrease in the concentration of antioxidants, such as carotenoids, ascorbic acid, vitamin E, vitamin A, and selenium characterize the depletion of liver hepatic reserves. As a consequence, there is an increase in lipid peroxidation (LPO) and protein carbonylation [11,12]. Furthermore, some studies suggest the presence of OS in the early stages of NAFL and in its progression to cirrhosis and, in a reduced number of cases, it leads to the predisposition to hepatocellular carcinoma [10].

Therefore, it is important to seek for strategies that improve the antioxidant capacity in subjects who suffer from NAFL as a consequence of MS. Here, we analyzed whether a mixture of RSV + QRC, in part, reverses the oxidative damage that is produced by high sucrose consumption in the liver in a MS rat model that develops NAFL.

## 2. Materials and Methods

### 2.1. Animals and Diet

The Laboratory Animal Care Committee of the National Institute of Cardiology “Ignacio Chávez” in Mexico (protocol #14-860) approved the experiments in animals and they were conducted in compliance with the Guide for the Care and use of laboratory animals of the National Institutes of Health (NIH). Weaning male Wistar rats weighing 50 ± 5 g were used (*n* = 16 male rats per group), and were randomly separated into two groups. Group 1, MS rats, receive 30% sugar in their drinking water for 20 weeks and group 2, control rats (C), which were given tap water for drinking. Half of each group of rats (MS or C) orally received sucrose solution or drinking water a mixture of RSV and QRC daily for four weeks in a dose 50 mg/kg/day–0.95 mg/kg/day, respectively (provided by ResVitalé TM, which contains 20 mg of QRC per 1050 mg of RSV). Groups without RSV + QRC treatment only received the vehicle. The mixture of RSV and QRC had been previously dissolved in 1 mL ethanolic solution (20%).

The rats were housed with a 12:12 light-dark cycle. The rodent commercial food that was provided to the animals contained: 23% crude protein, 4.5% crude fat, 8% ashes, and 2.5% added minerals (Lab Diet 5001; Richmond, IN, USA) and was given at libitum. At the end of the treatment, the animals were weighed and their SBP was measured by the tail-cuff method, as previously described [6]. After overnight fasting, the rats were subjected to euthanasia with a guillotine. The intra-abdominal fat was separated, weighed, and then discarded. The livers were excised, weighed, and divided for histological and biochemical analyses while fresh.

### 2.2. Serum Biochemistry Parameters

The blood sample was collected from fasting rats (12 h) and then centrifuged for 20 min at 644 rcf, at 4 °C. The serum was separated and stored at −30 °C, until the analysis of blood chemistry was performed. Cholesterol and TG determinations were made using commercial enzymatic kits (RANDOX Laboratories Ltd., Crumlin, County Antrim, UK). Insulin, leptin, and adiponectin were determined using commercial radioimmunoassay kits (RIA) (Linco Research Inc., Saint Charles, MI, USA). Glucose concentration was determined by enzymatic kit SERA-PAKR Plus (Bayer Corporation, S´ees, France). IR was estimated from the homeostasis model (HOMA-IR) and calculated from insulin levels and the fasting glucose, according to the formula: (insulin (μU/mL) × glucose (mmol/L)/22.5).

### 2.3. Lipid Detection in Fatty Liver

The left lobe of liver was excised immediately after animals were euthanized; 5 mm of liver tissue were fixed with buffered formalin (4%) supplemented with NaCl (0.09%). Tissue 10 μm cryo-sections were obtained and mounted on gelatinized slides and then kept refrigerated until staining. Rat liver cryo-sections were stained with oily red (OR) (Sigma-Aldhich-O0625). For staining, the cuts were dehydrated in 60% isopropanol (RA: IT2185-4) and incubated for 5 min in a solution of OR (previously prepared to saturation with isopropanol). Excess dye was removed with a quick wash with isopropanol and two washes with distilled water, and subsequently the tissue was mounted with aqueous medium. Observation was made under a microscope (FLoid® Cell Imaging Station, Life Technologies, Carlsbad, CA, USA) and photographs were obtained. The images were obtained with relief phase and red fluorescence channels in a transmitted light (TL) station. The ImagePro Premier calculated the arbitrary units of fluorescence (AUF). No mark was detected in the liver cryo-sections that were processed for the staining, but without OR, in any of the experimental groups (negative controls).

### 2.4. Liver Homogenization

The livers were homogenized in sucrose solution (sucrose 25 mM, Tris 10 mM and EDTA 1 mM, pH 7.35) with protease inhibitors (pepstatin 2 μM, leupeptin 2 μM, PMSF 1 mM, and 0.1% aprotinine) and then centrifuged at 1789 rcf for 10 min at 4 °C. The supernatant was separated and stored at −30 °C. The Bradford method determined the total proteins [13].

### 2.5. Evaluation of Antioxidant Enzymatic System in Native Gels

The measurement of the activity of antioxidant enzymes CAT, SOD isoforms, and peroxidases was performed by electrophoresis in 10% polyacrylamide native gels. CAT and SOD isoforms were revealed according to the methods that were previously described by Pérez-Torres [14], and peroxidases were revealed by the method that was described by Fridovich [15]. Purified SOD from bovine erythrocytes, with a specific activity of 112 U/mg of protein (Sigma-Aldrich, St. Louis, MO, USA), purified CAT from bovine liver having a specific activity of 60 U/mg (Sigma-Aldrich), and horseradish peroxidase with a specific activity is 61 U/mg of protein (from Sigma-Aldrich), were used as positive controls. The activity of the antioxidant enzymes previously mentioned, which the manufacturer provided, was placed in a separate lane of the gel and run in parallel with the biological samples. The intensity of the signal from the controls was used as a reference to measure the enzymatic activity in the tissue samples. Therefore, the results are expressed as U of activity per mg of protein. The gels were analyzed by densitometry with image Sigma Scan Pro 5.1 (Systat Software, Inc., San Jose, CA, USA).

### 2.6. Glutathione-S-Transferase and Glutathione Reductase Activities

The method of Beutler determined the glutathione-*S*-transferase (GST) activity in 1 mg of protein from the liver homogenate [16]. Upon conjugation of the thiol group of glutathione to the CDNB substrate, there is an increase in the absorbance at 340 nm. The rate of increase in the absorption is directly proportional to the activity of GST in the sample. The GST activity is expressed in µM of CDNB-GSH conjugate formed/min/mg protein. The glutathione reductase (GR) activity, in 1 mg of protein from the liver homogenate, was determined by the method that was previously described by Soto [17], and the absorbance was read at 340 nm. GR activity is expressed in µmol of reduced GSSG/min/mg protein.

### 2.7. Evaluation of Oxidative Stress Markers

#### 2.7.1. Protein Carbonylation

To measure the carbonylation of the proteins, 1 mg of protein from liver homogenate was added to 500 μL of HCl 2.5 N or 500 μL of 2,4-dinitrophenylhydrazine (DNPH). It was vigorously mixed with a vortex and then incubated in darkness at room temperature, for one hour, shaking with the vortex every 15 min. At the end of the incubation period, 500 μL of trichloroacetic acid 20% were added, and the mixture was then centrifuged at 16,099 rfc for 5 min. The supernatant was discarded, continuing with the washings, by first removing the precipitate with a sealed capillary tube, adding 1 mL ethanol/ethylacetate, allowing 10 min rest between each washing to permit the elimination of DNPH, and then mixing and centrifuging at 16,099 rfc for 10 min. This was repeated twice. Finally, 1 mL of solution of guanidine hydrochloride 6M with monobasic phosphate 20 mM pH 2.3 was added and then incubated at 37 °C for 30 min. Absorbance was read in a spectrophotometer at 370 nm, using air as blank and molar absorption coefficient of 22,000 M^−1^ cm^−1^ [18].

#### 2.7.2. Lipid Peroxidation

Lipid peroxidation (LPO), which is a marker of unsaturated cell membrane lipid damage by free radicals, in 1 mg of protein from the liver homogenate, was measured by the TBARS method that was previously reported [18]. In this assays, the end product malondialdehyde (MDA), which is a reactive aldehyde produced by lipid peroxidation of polyunsaturated fatty acids that forms a 1:2 adduct with thiobarbituric acid, is determined. The absorbance at 532 nm is proportional to the concentration of MDA and the calibration curve was obtained using tetraethoxypropane as a standard.

#### 2.7.3. The Total Antioxidant Capacity (TAC)

The total antioxidant capacity (TAC) of the non-enzymatic system in 1 mg of protein from the liver homogenate was determined according to the method that was previously described by Soto [18]. The absorbance was measured at 593 nm. The calibration curve was obtained using Trolox, which is a water-soluble antioxidant and it is commonly used as a positive control in antioxidant assays.

#### 2.7.4. Quantification of GSH Levels

The Ellman reactive (5,5′-dithiobis-2-nitrobenzoic) made the determination of GSH levels, according to the method described by Pérez-Torres [14]. The absorbance was read at 412 nm. The calibration curve was performed using GSH at concentrations ranging from 5 to 25 µmol/mL (positive control).

### 2.8. Western Blotting of Nrf2

Liver protein extracts (50 μg) were run on 12% SDS-PAGE (bis-acrilamide-laemmli gel) and blotted onto a polyvinylidene difluoride (PVDF) membrane (0.22 μm Millipore, Billerica, MA, USA) and then blocked 1 h at room temperature with Tris buffer solution-0.01% Tween (TBS-T 0.01%) plus 5% non-fat milk. The membranes were incubated overnight at 4°C with mouse primary monoclonal antibody Nrf2 (sc-365949) from Santa Cruz Biotechnology, Santa Cruz, CA, USA, at a final dilution of 1:1000. After that, the membranes were incubated overnight at 4 °C with a secondary antibody that is conjugated with horseradish peroxidase and a dilution 1:10,000 (Santa Cruz Biotechnology, Santa Cruz, CA, USA). All of the blots were incubated with α-actin antibody as control. The protein was detected by chemiluminescence assay (Clarity Western ECL Substrate, Bio-Rad Laboratories, Inc., Hercules, CA, USA). Chemiluminescence that was emitted in this process was detected in X-ray films (AGFA, Ortho CP-GU, Agfa HealthCare NV, Belgium). Images from each film were acquired with a GS-800 densitometer (including Quantity One software from Bio-Rad). The values of the density of each band are expressed as arbitrary units (AU).

### 2.9. Statistical Analysis

The analysis of the data to determine whether the differences between the means obtained were statistically significant was made using one-way ANOVA and a Tukey test with a significance level of *p* < 0.05. The data are shown as mean ± standard error. This analysis and graphs were made with the program Sigma Plot version 12.3, Jandel Corporation (version 2016, Systat Software Inc., San Jose, CA, USA).

## 3. Results

### General Characteristics and Serum Biochemical Measurements.

Table 1 shows the general characteristics of the experimental groups. Body weight, intra-abdominal fat, and SBP were significantly increased in the MS group in comparison to C group, and the RSV + QRC treatment restored the variables to levels that were similar to those in the C group (*p* = 0.001). Our results revealed no significant difference in the relative liver weights in all groups. Table 1, which also shows that TG, insulin, HOMA index, leptin, and adiponectin were significantly increase in MS rats when compared to C animals (*p* = 0.001). In MS rats, the treatment with RSV + QRC prevented the increase in body weight and significantly decreased the central adiposity. RSV + QRC significantly reduced the insulin concentration in MS rats and restored HOMA-IR. The mixture of polyphenols also significantly increased insulin sensitivity in the C group (*p* = 0.05).

Leptin and adiponectin concentrations showed a significant increase in the MS group when compared to the C group (*p* = 0.001). The treatment with RSV + QRC decreased the leptin levels in the MS group in comparison to the MS group without treatment (*p* = 0.05, Table 1); however, the concentrations of this adipokine remained high when compared to controls. A tendency towards lower values was observed with the high dose in adiponectin concentrations. The fasting serum glucose and cholesterol concentrations were not significantly different among the groups.

#### Evaluation of Oxidative Stress Markers

Table 2 describes that carbonylation in the MS plus RSV + QRC group was significantly decreased in comparison to the MS group (*p* = 0.001). Table 2 also shows that LPO was significantly lower in the C and MS plus RSV + QRC groups in relation to the MS group (*p* = 0.01 and *p* = 0.001, respectively).

The TAC of the non-enzymatic system showed a significant increase in the C and MS plus RSV + QRC groups in comparison with the MS group (*p* = 0.04 and *p* = 0.001, respectively, Table 2).

The GSH concentration showed a similar tendency than that of the TAC, in C and MS plus RSV + QRC groups vs. MS group (*p* = 0.04, Table 2).

#### Histopathological Findings

The micrographs that used oily red (OR) staining in rat liver slices showed small fat drops, which were located around the hepatic vein in the liver of the C group; however, in the liver of the MS rats, there were more fat drops than in the C group (reaching up to 5.86 times; *p* < 0.001). The RSV + QRC treatment in both MS and C groups showed a significant decrease in comparison to the MS and C groups (*p* < 0.001, Figure 1 and Figure 2).

#### Antioxidant Enzymatic Activities

Figure 3A shows that the Mn-SOD activity in the liver homogenate was increased in C and MS plus RSV + QRC groups vs. MS group (*p* = 0.02 and *p* = 0.01, respectively). A similar tendency was observed with the isoform Cu/Zn-SOD; this enzyme was increased in C and MS plus RSV + QRC groups vs. MS group (*p* = 0.01 and *p* = 0.01, respectively, Figure 3B).

The CAT activity in the liver homogenate showed a significant decrease in the MS group when compared to that in the C group (*p* = 0.03). The treatment with RSV + QRC in MS rats did not modify the activity of this enzyme (Figure 4).

Figure 5 shows that the peroxidase activity in the liver homogenate was increased in the C and MS plus RSV + QRC groups vs. MS group (*p* = 0.001).

Figure 6A shows that the GST activity was increased in C and MS plus RSV + QRC groups vs. MS group (*p* = 0.01 and *p* = 0.01, respectively).

The GR activity was decreased in the liver homogenate of the MS rats in comparison to C rats (*p* = 0.03); however, the treatment with RSV + QRC restored it (*p* = 0.01, Figure 6B).

We also evaluated the effect of RSV + QRC on the Nrf2 expression, which constitutes one of the main regulators of the set of antioxidant genes. The levels of Nrf2 tend to decrease without reaching significant difference in the MS rats in comparison to C group However, in MS plus RSV + QRC group, the treatment increased the Nrf2 expression (*p* = 0.001, Figure 7).

## 4. Discussion

The liver is an organ in which there are anatomical and physiological alterations due to the presence of the signs that comprise the MS. One of these alterations is fatty liver and, in this condition, the organ shows inflammation, fibrosis, ROS accumulation, OS, LPO, carbonylation, and changes in the activities of the antioxidant enzymes [19]. A better understanding regarding the pathogenesis of fatty liver in the MS and of the underlying OS that causes it is needed to develop new possible treatments. The polyphenols and flavonoids, from diverse vegetables and fruits, may be utilized as potential treatments for the signs that comprise the MS. Among the individual polyphenols and flavonoids that have been examined, RSV and QRC have been reported to have benefic effects separately one from the other. However, the reports on the effects of the combination of these natural compounds that are currently commercially available and have antioxidant properties are scarce.

### 4.1. Effect of RSV and QRC on Altered Biochemical Variables and Associated Signs in MS

In our study, the commercial combination of RSV + QRC improved the parameters of MS induced by the administration of 30% sucrose in drinking water for 24 weeks since weaning in rats (body weight, intra-abdominal fat, TG, insulin, HOMA-index, leptin and adiponectin concentrations in serum). We consider that not having determined the flavonoid and polyphenol residual content in the commercial food that is administered to our experimental animals and not being able to determine what proportion of the flavonoids and polyphenols were administered by the treatment with respect to their total intake constitutes a limitation of the present study. The residual content of these compounds in the diet might be important and it could alter the evaluated variables. Nevertheless, there were no significant changes in the food intake of the studied groups (MS RSV + QRC = 12.8 ± 0.7 g/day per rat, MS = 14.6 ± 0.8 g/day per rat; C = 19.2 ± 0.5 g/day per rat; C RSV + QRC = 17.4 ± 1 g/day per rat) and therefore we believe that the effect observed was mainly due to the administration of the RSV and QRC mixture in the drinking water. 

In some studies, in rodents and humans, RSV administration (5 and 25 mg/kg) leads to a decrease in the variables that are mentioned above; these results are consistent with the findings of the present study [20]. It has been reported that there is an inverse relationship between circulating adiponectin levels and adipose tissue mass, especially with central adiposity. However, we found that, even in the presence of high serum adiponectin levels, the MS rats showed dyslipidemia. Paradoxical results in mice and in human populations have been reported regarding high adiponectin levels and dyslipidemia. Some authors have suggested a high adiponectin concentration, together with a greater risk of cardiovascular diseases, obesity, and IR, while others have found no association between adiponectin and MS [21,22]. Future studies that are aimed at elucidating the mechanisms that underlie these results are needed; but, one possible explanation to this discrepancy is that MS animals might have adiponectin resistance. Nevertheless, the treatment with natural compounds decreases leptin resistance and high adiponectin, and this could be associated to intra-abdominal fat and IR reductions [23].

Increased central adiposity characterize MS and RSV and QRC can prevent it. However, there are reports showing conflicting data regarding the effect of RSV treatment on body weight. In our study, RSV induced not only the normalization of body weight, but also a decrease of intra-abdominal fat. A possible mechanism for this effect might be the activation of SIRT1; our group previously showed that the RSV + QRC mixture reduces hypertension, IR, body fat, and dyslipidemia in the MS rat model, and these changes were associated with the overexpression of PPARγ, SIRT-1, and SIRT2 in intra-abdominal adipose tissue [24]. The effect of the mixture could be mimicking the activation of the same pathway that is activated by calorie restriction. RSV is thought to interact with SIRT1 to activate peroxisome proliferator-activated receptor gamma coactivator 1α, which increases the use of cellular energy stores in brown adipose tissue and skeletal muscle [25]. Furthermore, it has been reported that RSV can modulate the uncoupling of ATP production from the electron transport chain across the membrane of the mitochondria in adipose tissue, ultimately burning calories by thermogenesis [26]. It has been described that QRC can decrease TG in obese rats [20]. QRC is also effective in decreasing adipogenesis in preadipocytes. In contrast, RSV only acts on mature adipocytes. Therefore, the RSV + QRC mixture may have synergic effects on SIRT1 and SIRT2, which contribute to decreasing TG, body weight, and intra-abdominal fat [6].

IR also characterizes MS. In our experimental model of MS, the insulin levels were increased, and the RSV + QRC treatment prevented their elevation. The HOMA-index, which indicates IR, showed the same tendency. These results are in agreement with a previous study that reported that RSV could improve insulin sensitivity and metabolic complications in fructose-fed rats [27]. Another study had suggested that RSV might decrease IR in MS rats through the regulation of the rennin- angiotensin system [28]. RSV can influence a number of pathways that regulate insulin metabolism in the liver. Insulin receptor substrate 1 (IRS-1) is a key regulator of insulin signaling. When stimulated by insulin, it promotes glycogen synthesis and inhibits glucose release from the liver. IRS-1 also stimulates the phosphoinositide-3-kinase pathway, which, in turn, phosphorylates AKT at T308, and also by the mTOR-rapamycin-insensitive companion of mammalian target of rapamycin complex at a second activation site, S473 [29]. The activation of these pathways can enhance the synthesis of glycogen and improve insulin sensitivity, leading to a decline in IR [30]. A previous study reported that RSV may up-regulate SIRT1 protein in the liver and, in consequence, activate AMPK in KKA(y) mice [31].

Another of the pathologies that are involved in MS is hypertension. QRC lowers SBP through the activation of endothelial nitric oxide synthase (eNOS). Our results show that SBP was decreased by the RSV + QRC treatment in MS group to levels that are similar to those found in the C group. The explanation of this finding may involve the increase in the expression or activity of eNOS [32]. A study reported that RSV up-regulates the expression of this enzyme and it consequently elevates nitric oxide (NO) synthesis [33]. NO improves endothelial function and thus decreases SBP, as shown in our results. Besides, an improvement in insulin sensitivity can lead to increased eNOS expression and its phosphorylation [34].

ROS overproduction also leads to oxidation of tetrahydrobiopterin (BH4), which is the essential cofactor of eNOS. In BH4 deficiency, the ROS uncouple NOS activity, thereby converting the enzyme into a superoxide-producing protein. Consequently, NO production is reduced, and the pre-existing OS is enhanced, thus favoring hypertension that is associated to this disease [35]. QRC promotes the relaxation of vascular smooth muscle through an increase of eNOS expression and contributes to the decrease of SBP, as shown in our results [36]. In addition, QRC mediates an increase in Ca^2+^ in cultured endothelial cells with the subsequent stimulation of eNOS phosphorylation [37]. Moreover, in porcine coronary arteries, QRC inhibits the PI3K pathway and potentiates relaxation through eNOS activation [38]. Thus, our results suggest that the treatment with the mixture the RSV + QRC may have a synergic effect on the reduction of obesity, IR, and hypertension in MS rats.

### 4.2. Effect of RSV and QRC on NAFL and on Hepatic OS

Our results show that the treatment with the mixture of RSV + QRC significantly decreased the lipid deposition in the liver in rats with MS, thus contributing to restoring TG levels, increasing insulin sensitivity to levels that are similar to those found in C rats. This observation was evidenced in the histological sections (Figure 1) and it is consistent with other studies [29].

IR leads to hepatocellular fat deposition and liver damage, including NAFL and cirrhosis [39]. In studies that were done in small-animals, RSV could restore insulin sensitivity and signaling in the liver and facilitate reverse transport of cholesterol, preventing the lipid dysfunctional metabolism [25]. In another study, RSV treatment reduced hepatic lipid accumulation to normo-cholesterolemic control levels [40]. The metabolic alterations that are involved in MS increase hepatic OS [41]. RSV and QRC improve the antioxidant capacity and exert a protective effect against OS in the fatty liver in several experimental models, such as the ethanol- inducing model and in the streptozotocin-induced diabetic rat model [41,42,43].

The liver plays an important role in the detoxifying process of free radical and in maintaining OS biomarker levels that are exacerbated in fatty liver disease. LPO and protein carbonyls are secondary products of OS and they have, as a consequence, toxic damage for the hepatic cells. The high levels of these oxidative products are indexes of the degree of hepatocyte damage [44]. The GST enzyme that is GSH dependent is important for the detoxification of these oxidation products. GST participates in the cellular detoxification and excretion of many physiologic and xenobiotic substances. GST may inactivate endogenous aldehydes, quinones hydroperoxides, and epoxides that are formed as secondary metabolites from the OS [45]; QRC and RSV may increase the GST activity in rats with NAFL [46].

The results of our study show that TAC and GSH were decreased and that there was an increase in LPO levels and protein carbonylation in the fatty liver of MS rats. The administration of the mixture of RSV + QRC prevented this damage and favored the GST activity. This finding suggests that the mixture of polyphenols and flavonoids could have a protective effect against hepatic damage. The suppressive effect of these natural compounds on the oxidation of unsaturated fats in the cell membrane may underlie the decrease in LPO levels through GST activity. Furthermore, RSV has been reported to protect the liver of streptozotocin induced diabetic rats from LPO by [44]. LPO can generate a cascade of reactions that produce endogenous toxic substances that react with the adjacent membrane proteins or diffuse to more distant molecules, such as DNA. This leads to a positive feedback loop that causes anatomical anomalies and physiological complications in the fatty liver [47]. Additionally, LPO products can promote collagen production that leads to adducts with proteins that participate in the initiation of an inflammatory response that is associated to fibrosis in the NAFL [48].

Our results show that the commercial mixture of RSV + QRC may not only favor a lowering of LPO, by inhibiting the Fenton reaction, but it can also slow the production of free radicals. This effect can be, in part, attributed to the hydrogen electron donations from the OH groups that are present in both molecules [48], but also to an increase in the activity of the antioxidant enzymes. 

RSV + QRC improved the liver antioxidant defense in MS rats and lowered LPO and protein carbonylation. The activities of the SOD isoforms represent the first line of intracellular defense against cellular damage from ROS, followed by the activities of peroxidases and CAT. The results of this paper show an increase in the activities of both the SOD isoforms and of the peroxidases that are caused by the treatment with the mixture of RSV + QRC in MS animals (Figure 3 and Figure 5). The expression of CAT and SOD isoforms has been previously determined in the liver of streptozotocin induced diabetic rats and has been found to be increased after the administration of RSV [41]. In addition, another study showed that the treatment with RSV significant decreased LPO but increased GSH levels, GST, CAT, SOD, and quinone reductase in the liver of diabetic rats [47].

A previous study showed that the activity of CAT is higher in polyphenol consuming rodents than in the controls animals [44]. Although the activity of CAT is significantly diminished in MS rats, our results show that the RSV + QRC treatment did not significant modify CAT activity (Figure 4). CAT activity inhibition may be due to H2O2 overproduction. This enzyme depends on the concentration of the substrates, in contrast to the peroxidases, which do not depend on the substrate concentration, but on GSH and NADPH. This property of peroxidases renders them into more efficient antioxidant systems. Peroxidases may also detoxify the H2O2 into H2O and molecular O2 [49].

Furthermore, the depletion of the tripeptide GSH, which is a key modulator of cell functions, is associated with hepatotoxicity [50]. The modulator functions of GSH include its antioxidant defense, redox regulation of protein thiols, and maintenance of redox homeostasis, and it also has reducing and nucleophilic properties. GSH biosynthesis and recycling depends on several enzymes, including γ-glutamylcysteine synthetase (γ-GCS), GR, and glutamate-cysteine ligase (GCL), which are key enzymes in GSH homeostasis [51]. The GR is an enzyme that recycles the GSSG to GSH and consumes NADPH [49]. However, in an OS state, this pathway is diminished. Our results show that the RSV + QRC treatment significantly increased the GR activity, which may contribute to restoring the GSH levels in the liver of MS rats (Figure 6B). 

### 4.3. Effect of RSV + QRC on Nrf2

Flavonoids have been proposed to induce the expression of protective genes, in particular, those encoding enzymes that are involved in GSH synthesis, through the activation of the master transcription regulator nuclear factor erythroid-2 (Nrf2). The results of this paper show that the mixture of RSV + QRC may increase GSH synthesis or its regeneration, which contributes to the neutralization of ROS. This result is in agreement with a previous study [52]. In neuron cultures, QRC pretreatment for 24h with H2O2 increased the GCL expression and the total GSH levels. This was associated to the activation of the Nrf2 signaling pathway [53]. Furthermore, QRC 50 µM regulates GSH levels, employing antioxidant enzymes and Nrf2, and targeting p38-MAPK activation; this action might play an important role in the restoration of cellular redox homeostasis in the fatty liver [54]. In C6 astroglia cells, RSV can modulate the GSH system through the heme oxygenase 1 and Nrf2 pathways [55].

Our results show that Nrf2 expression showed a tendency to decrease in livers from MS rats when compared to C rats; and, the RSV + QRC treatment increased the levels of Nrf2 in MS rats (Figure 7). This suggests a synergic effect this mixture on the expression and activity of the antioxidant enzymes that participate in the reduction the ROS in fatty liver, such as SOD isoforms, peroxidases, GST, and in the enzymes that maintain the GSH recycling, such as GR. It has been reported that RSV treatment attenuates protein oxidation and improves SOD, GPx, CAT, GST, and GSH levels through the activation of Nrf2 mRNA levels and SIRT1 in diabetic rats and human liver carcinoma cells [56]. Under normal conditions, Nrf2 plays a central role in basal activity and in the coordinated induction of many genes, including those of the phase II antioxidant enzymes. Impaired Nrf2 activation may contribute to the development of the signs that are present in the MS, including inflammation, hypertension, and OS [57]. 

Furthermore, changes in cellular redox homeostasis can lead to Nrf2 activation. It then dissociates from keap1 and translocates to the cell nucleus, causing the acetylation of DNA, binding to the elements of the antioxidant response in the promoter region of the genes that are involved in the transcription of the already mentioned antioxidant enzymes [58].

## 5. Conclusions

The mixture of RSV + QRC has benefic effects on the OS that is present in the fatty liver in the MS rats through the over-expression of the master factor NrF2. NrF2 favors the increase of the antioxidant enzymes and of GSH recycling. At the same time, the administration of natural compounds increases the activity of antioxidant enzymes. These effects may be associated with a decrease in fat accumulation in the liver.

## Figures and Tables

**Figure 1 molecules-24-01297-f001:**
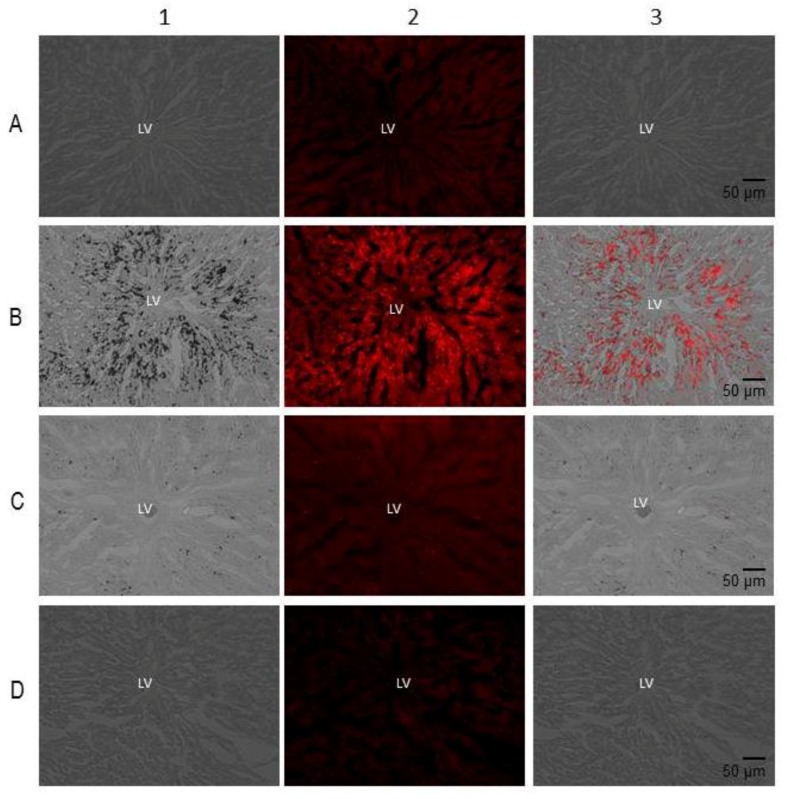
Representative micrographs of oily red staining in rat liver slices. The images were obtained using transmitted light. Abbreviations: MS = Metabolic syndrome, C = Control, RSV + QRC = resveratrol plus quercetin. Panel A = MS plus RSV + QRC, Panel B = MS, Panel C = C, Panel D = C plus RSV + QRC, LV = liver vein. Vertical line 1 = relief phase, vertical line 2 = red fluorescence channels merge and vertical line 3 = upper right end on the FLoid^®^ Cell Imaging Station.

**Figure 2 molecules-24-01297-f002:**
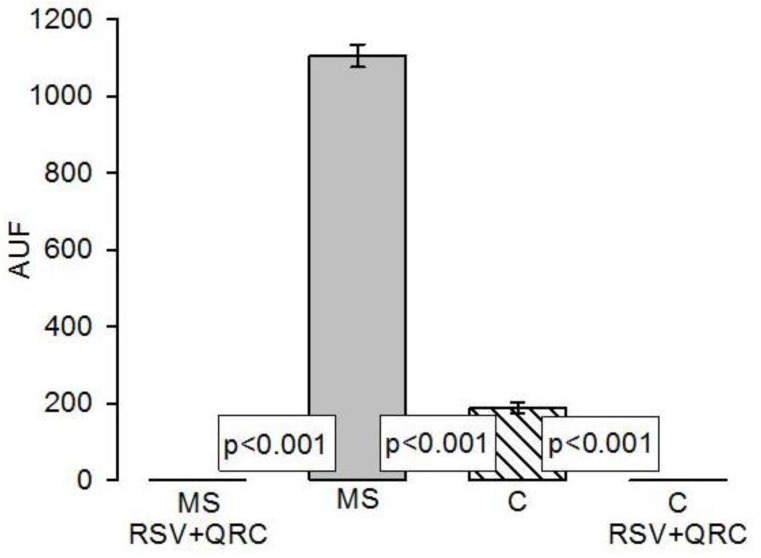
Histogram that represent the Lipid detection from rat liver cryosections stained with oily red (OR) and imaged using transmitted light (TL), contrast phase and red fluorescence channels on the FLoid^®^ Cell Imaging Station. Abbreviations: AUF Arbitrary units of fluorescence calculated with the image-pro program in an Olympus fluorescence microscopy.

**Figure 3 molecules-24-01297-f003:**
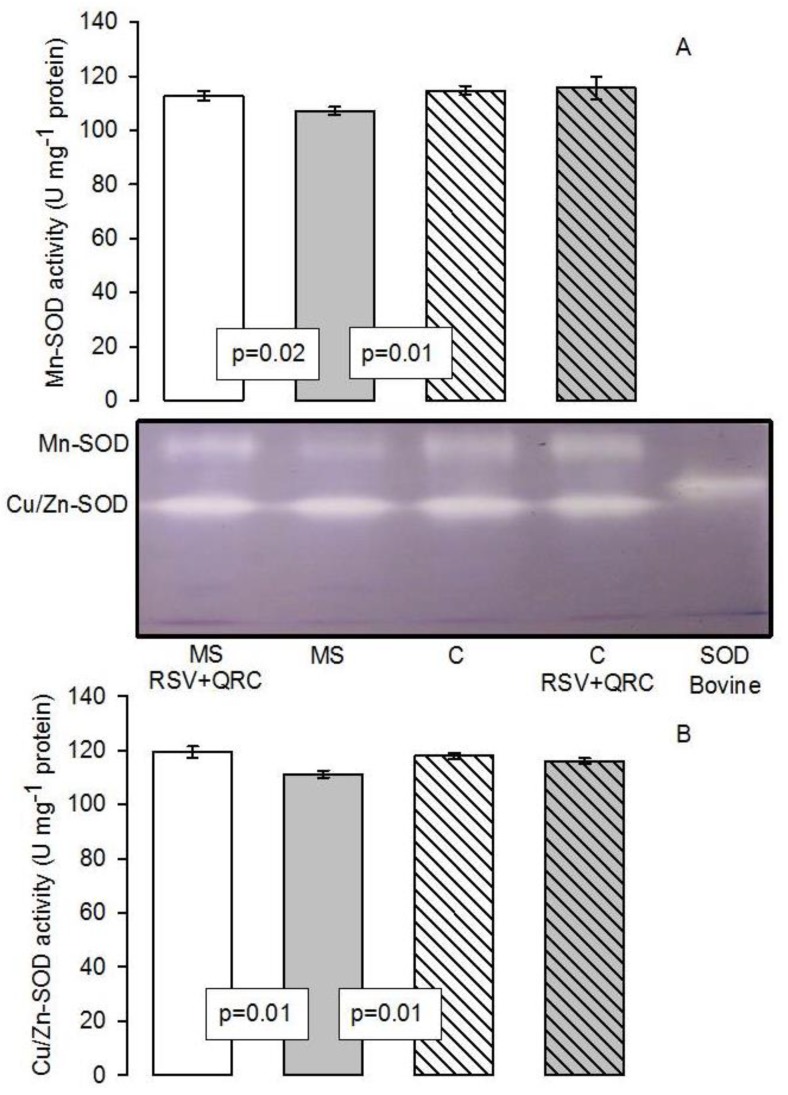
Histogram representing the densitophotometric analysis of the activity of superoxide dismutase (SOD) isoforms in the liver homogenates of the experimental rats. A representative native gel of the activities of the Mn-SOD and Cu/Zn-SOD electrophoresis between the graphic A and B is shown. (**A**) Manganese isoform, and (**B**) Copper/zinc isoform. Abbreviations: MS = metabolic syndrome, C = control, RSV = resveratrol, QRC = quercetin. Data are expressed in means ± SE, (*n* = 8, rats in each group).

**Figure 4 molecules-24-01297-f004:**
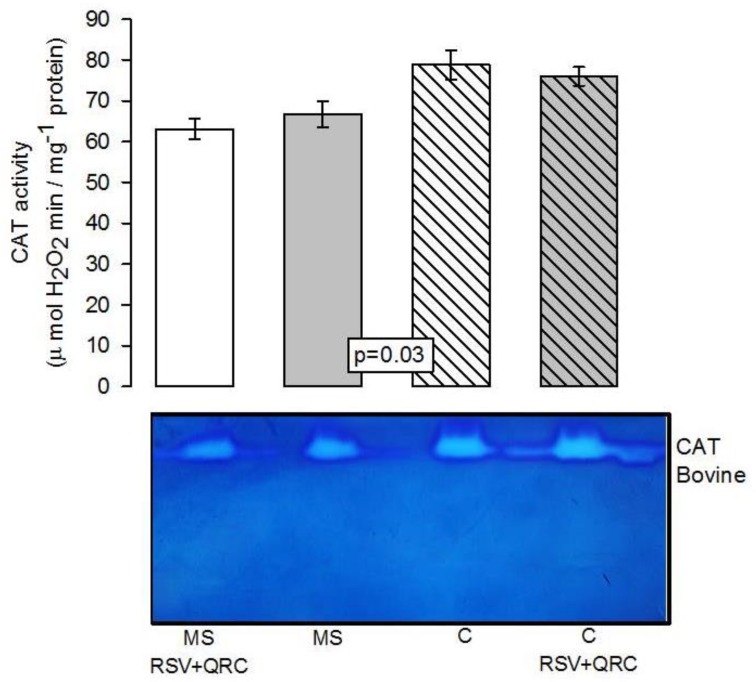
Densitophotometric analysis of catalase (CAT) activity in the liver homogenate in which the effect of treatment with the mixture of the RSV + QRC in experimental groups is shown. In the lower histogram, a native representative gel of the CAT activity is included. Abbreviations: MS = metabolic syndrome, C = control, RSV = resveratrol, QRC = quercetin. Data are expressed in means ± SE, (*n* = 8, rats in each group).

**Figure 5 molecules-24-01297-f005:**
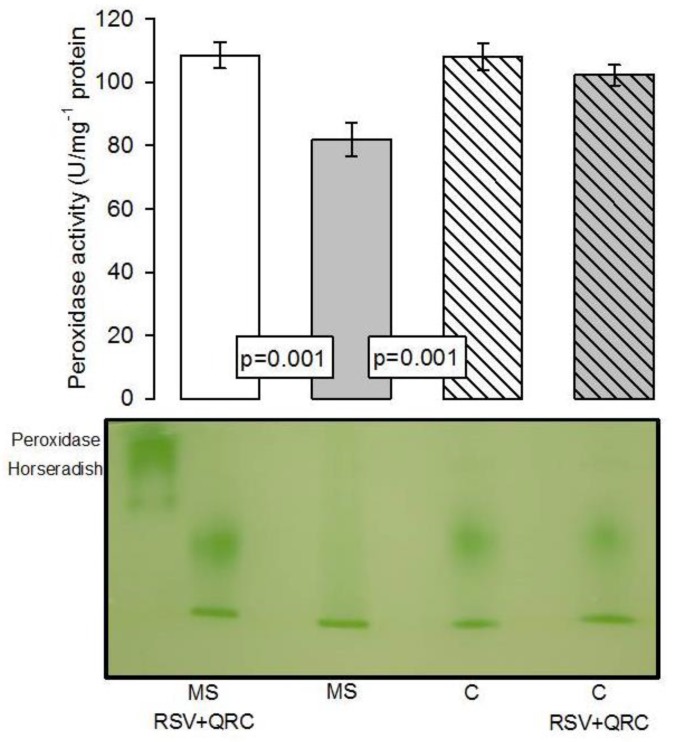
Densitophotometric analysis of the activities of peroxidases. A representative native gel is shown below the histogram. Under these conditions, where peroxidases are present, the gel remains transparent and the 3,3′,5,5′-tetramethylbenzidine is oxidized, showing a green coloration. Data are means ± SE, *n* = 8 each group. Abbreviations: MS = metabolic syndrome, C = control, RSV = resveratrol, QRC = quercetin.

**Figure 6 molecules-24-01297-f006:**
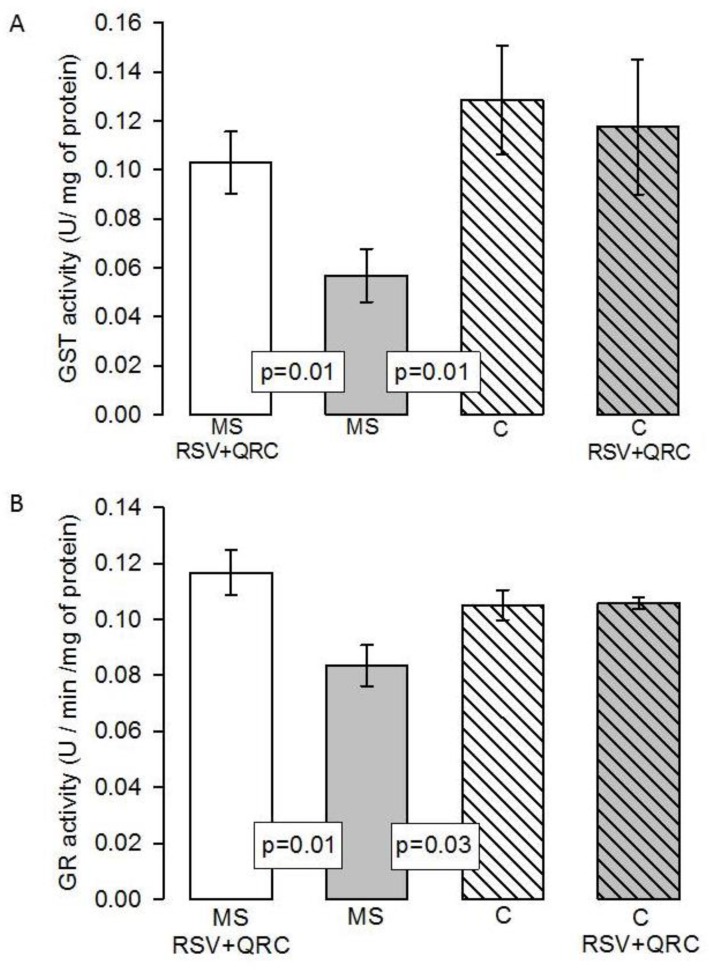
Effect of the treatment with natural compounds on Glutathione-*S*-transferase activity (**A**) and Glutathione Reductase (**B**) activity in liver homogenate. Data are means ± SE, *n* = 8 each group. Abbreviations: MS = metabolic syndrome, C = control, RSV = resveratrol, QRC = quercetin.

**Figure 7 molecules-24-01297-f007:**
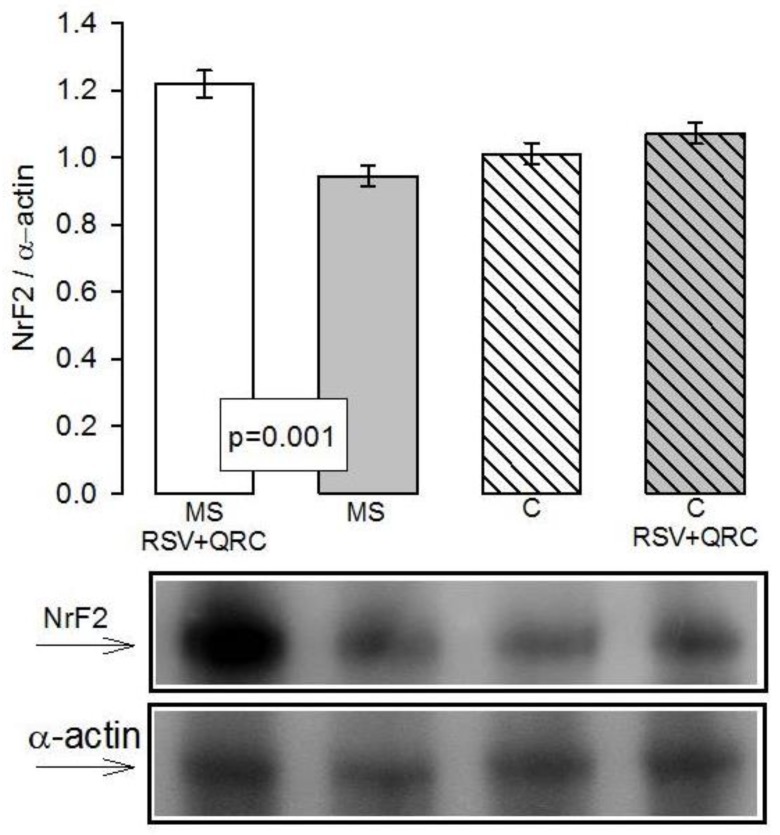
Effect of polyphenols on Nrf2 protein expression in liver homogenate from Control and MS experimental groups. A representative Western blot result is shown above the graphs. Data are means ± SE, *n* = 8 each group. Abbreviations: MS = metabolic syndrome, C = control, RSV = resveratrol, QRC = quercetin.

**Table 1 molecules-24-01297-t001:** General characteristics and serum biochemical in experimental groups.

	MS RSV + QRC	MS	C	C RSV + QRC
Weight (g)	481.8 ± 7.7	598.1 ± 9.1 ^b^	457.2 ± 13.7	514.2 ± 20.9
Intra-abdominal fat (g)	8.9 ± 1.0	12.9 ± 0.3 ^b^	4.9 ± 0.3	6.3 ± 0.9
Relative liver weight	2.6±0.1	2.4±0.05	2.4±0.06	2.3±0.04
SBP (mmHg)	115.4 ± 2.9	141.8 ± 0.9 ^b^	102.8 ± 0.8	110.8 ± 3.9
Glucose (mg/dL)	90.1 ± 2.2	95.5 ± 2.4	93.0 ± 3.3	91.1 ± 5.2
Cholesterol (mg/dL)	61.3 ± 1.5	63.5 ± 3.2	53.7 ± 2.9	60.3 ± 2.8
Triglycerides (mg/dL)	96.1 ± 4.6	141.5 ± 4.2 ^b^	72.5 ± 5.1	85.2 ± 8.3
Insulin (µU/mL)	0.16 ± 0.01	0.45 ± 0.05 ^b^	0.16 ± 0.03	0.11 ± 0.02
HOMA index	0.60 ± 0.07	2.16 ± 0.30 ^b^	0.81 ± 0.19	0.46 ± 0.10 ^a^
Leptin (ng/dL)	3.9 ± 0.1 ^a^	4.8 ± 0.3 ^b^	2.3 ± 0.2	2.5 ± 0.1
Adiponectin (μg/mL)	5.5 ± 0.1	7.5 ± 0.4 ^b^	3.7 ± 0.1	3.4 ± 0.2

Data are means ± SE, *n* = 8 each group. Statistically significant at ^a^ MS plus resveratrol (RSV) + quercetin (QRC) vs. metabolic syndrome (MS) and C vs. C plus RSV + QRC *p* = 0.05 and C vs. C plus RSV + QRC, ^b^ MS vs. C and MS plus RSV + QRC *p* = 0.001. The letters (superscripts) indicate the statistical significance. Abbreviations: MS=metabolic syndrome, C = control, RSV = resveratrol, QRC = quercetin, SBP = systolic blood pressure.

**Table 2 molecules-24-01297-t002:** Effect of Resveratrol and Quercetin administration on oxidative stress markers.

mg of Protein	MS RSV + QRC	MS	C	C RSV + QRC
Carbonylation (ng of carbonyls)	1.2 ± 0.2	2.2 ± 0.2 ^c^	1.8 ± 0.1	1.6 ± 0.3
LPO (nmol of MDA)	0.3 ± 0.1	0.9 ± 0.1c	0.4 ± 0.1 ^b^	0.5 ± 0.1
TAC (nmol of trolox)	675.6 ± 23.6	592.0 ± 29.5 ^a^	741.7 ± 19.9 ^d^	689.0 ± 17.3
GSH (nM)	4.1 ± 0.026	3.1 ± 0.043 ^a^	4.9 ± 0.066	4.2 ± 0.029

Data are means ± SE, *n* = 8 each group. Statistically significant at ^a^ MS vs. C and MS plus RSV + QRC *p* = 0.04, ^b^ C vs. MS *p* = 0.01, ^c^ MS vs. MS plus RSV + QRC *p* = 0.001 dMS vs. C *p* = 0.001. The letters (superscripts) indicate the statistical significance. Abbreviations: MS = metabolic syndrome, C = control, LPO = lipid peroxidation, MDA = malondialdehyde, TAC = total antioxidant capacity, GSH = reduced glutathione, RSV = resveratrol, QRC = quercetin.

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
