# Peer review of "Resveratrol and Quercetin Administration Improves Antioxidant DEFENSES and reduces Fatty Liver in Metabolic Syndrome Rats"

_molecules, 2019, doi:10.3390/molecules24071297_

Reviewer 1 Report

Introduction

References 5, 6 and 7 doesn´t match with what is written in the text. Authors must revise this fact and put references that support this part of the introduction.

 The sentence written from line 72 to line 73 doesn´t match very well with the rest of the paragraph (it should be changed/eliminated by the authors).

Material and methods

This section should be included after the introduction and prior to the discussion sections. On its current place (between the discussion and the conclusion sections is strange).

It is not clearly explained which was the dose of RSV+Q that the animals consumed. According to the authors, the mixture was prepared in 1 ml of ethanol ,and this was diluted in the drinkin water. My question is which was the RSV+Q dose that each animal was consuming during the study? The authors must explain better this part.

It is not detailed if the animals in the control groups (groups 2 and 4) received the vehicle (ethanol) in the drinking water.

Line 378: -30ºC.

It is not mentioned that any western blot analysis was carried out in this section. By contrast, in figure 7 a representative western blot is included. The authors must include the description of the procedure in the material and methods section.

Results

The sentence in lines 97 and 98 doesn´t make much sense (it should be revised).

Result included in table 1 should have letters instead of stars to better understand the results. On its current status, the results included in the table are difficult to understand/interpret. The same comment for table 2.

According to the reported results, it seems strange that no significant differences were found between C and C+RSV+Q groups regarding final body weights. Similarly, the results suggest that TG, adiponectin and leptin levels in the MS+RSV+Q group were significantly higher than in the C group. The authors should better explain those results.

Discussion

Overall, taking into account the reported results, the discussion is highly speculative. Indeed the authors talk about synergic effects of the combination of RSV and Q. Since these compounds were not used separately, these statements can´t be made.

Suggestions

English corrections must be done (several mistakes throughout the whole manuscript). Indeed, in some cases too informal language is used.

Author Response

Comments and Suggestions for Authors

Introduction

References 5, 6 and 7 doesn´t match with what is written in the text. Authors must revise this fact and put references that support this part of the introduction.

R= References were checked and now support this part of the introduction.

 The sentence written from line 72 to line 73 doesn´t match very well with the rest of the paragraph (it should be changed/eliminated by the authors).

R= The sentence was rephrased

Material and methods

This section should be included after the introduction and prior to the discussion sections. On its current place (between the discussion and the conclusion sections is strange).

R=Corrected

It is not clearly explained which was the dose of RSV+Q that the animals consumed. According to the authors, the mixture was prepared in 1 ml of ethanol, and this was diluted in the drinkin water. My question is which was the RSV+Q dose that each animal was consuming during the study? The authors must explain better this part. It is not detailed if the animals in the control groups (groups 2 and 4) received the vehicle (ethanol) in the drinking water.

R= The methods section was re-phrased to better explain the dose of RSV-QRC that the animals received.

Line 378: -30ºC.

R= Corrected

It is not mentioned that any western blot analysis was carried out in this section. By contrast, in figure 7 a representative western blot is included. The authors must include the description of the procedure in the material and methods section.

R= The description of the western blot analysis was included.

Results

The sentence in lines 97 and 98 doesn´t make much sense (it should be revised).

R=Corrected

Result included in table 1 should have letters instead of stars to better understand the results. On its current status, the results included in the table are difficult to understand/interpret. The same comment for table 2.

R= Stars were replaced by letters.

According to the reported results, it seems strange that no significant differences were found between C and C+RSV+Q groups regarding final body weights. Similarly, the results suggest that TG, adiponectin and leptin levels in the MS+RSV+Q group were significantly higher than in the C group. The authors should better explain those results.

Discussion

Overall, taking into account the reported results, the discussion is highly speculative. Indeed the authors talk about synergic effects of the combination of RSV and Q. Since these compounds were not used separately, these statements can´t be made.

R= The term synergic was avoided

Suggestions

English corrections must be done (several mistakes throughout the whole manuscript). Indeed, in some cases too informal language is used.

R= English was corrected and informal language avoided

Reviewer 2 Report

The authors investigated the combined effect of resveratrol (R) and quercetin (Q) on a rat model of metabolic syndrome (MS). Body and fat weights, systolic blood pressure, and serum metabolic parameters were measured. The levels of oxidative stress markers (carbonylation and lipid peroxidation, antioxidant capacity, and GSH) were also measured in liver homogenates. Liver accumulation of lipid droplets was analyzed by OR staining.  Antioxidant enzymatic activities (SOD, CAT, peroxidase, GST, and GR) were measured in liver homogenates. Lastly, Nrf2 protein expression levels were determined by western blotting. The data showed that R+Q administration improve antioxidant capacities of liver of sucrose-fed rats, resulting Q+R reduces oxidative stress, lipid accumulation, and possibly associated inflammation and attenuates fatty liver diseases. There are few comments.

The authors showed that adiponectin levels in MS were significantly increased compared to those of C, and R+Q decreased the levels in MS (shown in Table 1). However, general consensus on the effect of adiponectin is that adiponectin levels are inversely related with MS and contribute to lowering IR, OS, and inflammation supporting the anti-diabetic and anti-atherosclerotic effect of adiponectin. A recent review paper supports a therapeutic potential of adiponectin and states that increasing adiponectin levels reduce the risk of development of MS (Ghadge AA et al., 2018; doi: 10.1016/j.cytogfr.2018.01.004.). This discrepancy should be explained in the paper.

The manuscript should be edited by a language professional to check spelling or grammar errors. The discussion is not necessarily long, please reduce it at least half of the current length.

The image quality of Figures 3,4,5, and 7 should be improved, and the colors can be discarded.  

Author Response

The authors showed that adiponectin levels in MS were significantly increased compared to those of C, and R+Q decreased the levels in MS (shown in Table 1). However, general consensus on the effect of adiponectin is that adiponectin levels are inversely related with MS and contribute to lowering IR, OS, and inflammation supporting the anti-diabetic and anti-atherosclerotic effect of adiponectin. A recent review paper supports a therapeutic potential of adiponectin and states that increasing adiponectin levels reduce the risk of development of MS (Ghadge AA et al., 2018; doi: 10.1016/j.cytogfr.2018.01.004.). This discrepancy should be explained in the paper.

R= We added a paragraph addressing this issue in the discussion and included the suggested reference (line 321, reference #22 ).

The manuscript should be edited by a language professional to check spelling or grammar errors.

R= English was corrected

The discussion is not necessarily long, please reduce it at least half of the current length.

R= Done The discussion was shorted in we were thought necessary

The image quality of Figures 3,4,5, and 7 should be improved, and the colors can be discarded.  

R= the representative photomicrographs are of high resolution captured with a photographic camera of 14 megapixels, the colors cannot be discarted, because they are the own natural colors, that each one of the reactions produces in each one of the enzymes. SOD gives a purple color in the gel, but where there is activity of the transparent enzyme. Catalase gives a strong blue color in the gel and lighter where the activity of the enzyme is found. The peroxidase gives a light green color in the gene and where there is more activity of the enzyme the green intensifies.

Reviewer 3 Report

#1: The discussion should be expanded and discuss better the antioxidant-hypothesis. The literature cited is quite old, and more recent data suggest that in particular polyphenols might not act as physico-chemical antioxidants, but by affecting metabolic pathways (as is measured in the manuscript). This reviewers thinks that it would be better to focus less on antioxidants and their purported beneficial effect, but rather on compounds.

#2 Results are presented well

#3 Methodology

- it would be helpful to have more detailed information on the actual composition of the intervention, and the authors should provide these information (preferably measured, or from supplier). It is also important to know what potential contaminants can be found in the intervention

- did the chow contain any flavonoids or polyphenols (e.g. from soy or plants)?

- there is virtually no information on quality assurance and negative/positive control for the methods described. There should be information on repeatability, precision and the risk of contamination/false positives.

Author Response

#1: The discussion should be expanded and discuss better the antioxidant-hypothesis. The literature cited is quite old, and more recent data suggest that in particular polyphenols might not act as physico-chemical antioxidants, but by affecting metabolic pathways (as is measured in the manuscript). This reviewers thinks that it would be better to focus less on antioxidants and their purported beneficial effect, but rather on compounds.

R= References were substituted by modern ones. As we determined many variables regarding oxidative stress, we believe, it is important to discuss these data.

#2 Results are presented well

Thanks you for your comment

#3 Methodology

- it would be helpful to have more detailed information on the actual composition of the intervention, and the authors should provide these information (preferably measured, or from supplier). It is also important to know what potential contaminants can be found in the intervention

R= The Methods section was re-phrased providing the information from the supplier

- did the chow contain any flavonoids or polyphenols (e.g. from soy or plants)?

R= The commercial diet provided to the rats contained: Ground corn, dehulled soybean meal, dried beet pulp, fish meal, ground oats, brewers dried yeast, cane molasses, dehydrated alfalfa meal, dried whey, wheat germ, porcine meat meal, wheat middlings, animal fat preserved with BHA, salt, calcium carbonate, choline chloride, cholecalciferol, vitamin A acetate, folic acid, pyridoxine hydrochloride, DL-methionine, thiamin mononitrate, calcium pantothenate, nicotinic acid, dl-alpha tocopheryl acetate, cyanocobalamin, riboflavin, ferrous sulfate, manganous oxide, zinc oxide, ferrous carbonate, copper sulfate, zinc sulfate, calcium iodate, cobalt carbonate.

Some of these ingredients might contain flavonoids, specially the soybean. Furthermore, some antioxidants such as vitamins (vitamin E) are also present in the pellets. However, we did not measure the flavonoid content in the diet and are unable to discard that there may be an effect due to it. Nevertheless, the same diet was given to the control and MS groups and the effects are thus mainly due to the extra polyphenols and flavonoids administered with the treatment.

- there is virtually no information on quality assurance and negative/positive control for the methods described. There should be information on repeatability, precision and the risk of contamination/false positives.

R= All of the techniques used to determine the antioxidant enzymes are specific since they have all been previously standardized and proven in our laboratory and they have already been previously reported in other articles. The enzyme activities determined by zymograms have been compared to the activities of commercially available purified enzymes. This has been specified in the methods section.

Round  2

Reviewer 1 Report

The suggestions made in the previous report have been taken into account.

In the conclusion section the word synergic still remains (line 463). As I pointed in the previous review report, since both compounds haven´t been tested separatedly, this statement CAN´T be done under these experimental conditions.

Author Response

REVIEWER 1

 English language and style

( ) Extensive editing of English language and style required
( ) Moderate English changes required
(x) English language and style are fine/minor spell check required. Language was checked
( ) I don't feel qualified to judge about the English language and style

The suggestions made in the previous report have been taken into account.

In the conclusion section the word synergic still remains (line 463). As I pointed in the previous review report, since both compounds haven´t been tested separatedly, this statement CAN´T be done under these experimental conditions.

R= Corrected.

Reviewer 3 Report

The authors have not addressed two key comments:

Quality assurance: the authors need to state precision and accuracy of their measures, results of positive controls etc.

Feed: the residual flavonoid content is quite important, as it would allow a better interpretation of the results. If residual content contributed 90% of total intake in the intervention group, results would have to be interpreted quite differently. Moreover, even residual amounts could affect gene expression etc.

Author Response

REVIEWER 3

 English language and style

( ) Extensive editing of English language and style required
( ) Moderate English changes required
(x) English language and style are fine/minor spell check required. Language was checked
( ) I don't feel qualified to judge about the English language and style

The authors have not addressed two key comments:

1. Quality assurance: the authors need to state precision and accuracy of their measures, results of positive controls etc.

R= We added an explanation of the positive controls used in the Methods section. Explanations added are indicated as new comments highlighted in red (changes from round 1 are indicated by comments in black). We would like to mention that all of the techniques used in the present paper are specific and have been previously standarized and reported by our group and that all assays generate reproducible results.

For the detection of lipids in the cryo sections we add: “In the liver cryo-sections from all experimental groups processed for the staining, but without OR, no mark was detected (negative controls)” (line 123).

2. Feed: the residual flavonoid content is quite important, as it would allow a better interpretation of the results. If residual content contributed 90% of total intake in the intervention group, results would have to be interpreted quite differently. Moreover, even residual amounts could affect gene expression etc.

R=The reviewer is correct in his/her observation that the flavonoid and polyphenol intake comming from the pellets could affect our results. We now mention that not having determined this content constitutes a limitation of our study. Nevertheless, since there were no differences in the food intake in the different groups (MS RSV+QRC=12.8±0.7 g/day per rat, MS=14.6±0.8 g/day per rat; C=19.2±0.5 g/day per rat; C RSV+QRC=17.4±1 g/day per rat) we consider that the observed effect in our results is mainly due to the administration of RSV and QRCin the drinking water.  This explanation was added in the discusion section (line 316).